# Psychological Stress Perceived by Pregnant Women in the Last Trimester of Pregnancy

**DOI:** 10.3390/ijerph19148315

**Published:** 2022-07-07

**Authors:** Anca Răchită, Gabriela Elena Strete, Laura Mihaela Suciu, Dana Valentina Ghiga, Andreea Sălcudean, Claudiu Mărginean

**Affiliations:** 1Doctoral School, “George Emil Palade” University of Medicine, Pharmacy, Sciences and Technology, 540139 Targu Mures, Romania; anca.rachita@umfst.ro; 2Department of Psychiatry, “George Emil Palade” University of Medicine, Pharmacy, Sciences and Technology, 540136 Targu Mures, Romania; 3Department of Obstetric and Ginecology Clinic II, “George Emil Palade” University of Medicine, Pharmacy, Sciences and Technology, 540136 Targu Mures, Romania; laura.suciu@umfst.ro (L.M.S.); claudiu.marginean@umfst.ro (C.M.); 4Department of Medical Scientific Research Methodology, “George Emil Palade” University of Medicine, Pharmacy, Sciences and Technology, 540136 Targu Mures, Romania; dana.ghiga@umfst.ro; 5Department of Ethics and Social Sciences, “George Emil Palade” University of Medicine, Pharmacy, Sciences and Technology, 540136 Targu Mures, Romania; andreea.salcudean@umfst.ro

**Keywords:** emotional stability, anxiety, stressors, pregnancy, irrationality

## Abstract

Pregnancy is characterized by changes in neuroendocrine, cardiovascular, and immune function. For this reason, pregnancy itself is perceived as a psychological “stress test”. Research to date has focused on stress exposure. The aim of the study was to evaluate the influence of associated factors on the level of stress experienced by pregnant patients. We conducted a prospective study that included 215 pregnant women in the third trimester of pregnancy, hospitalized in the Obstetrics and Gynecology Clinic II in Târgu-Mureș, between December 2019 and December 2021, who were evaluated by the ABS II scale. All patients included in the study filled in a questionnaire that included 76 questions/items, in which all the data necessary for the study were recorded. The results obtained from the study showed that pregnant women in urban areas (53.49%) are more vulnerable than those in rural areas (46.51%), being influenced by social and professional stressors, social determinants playing a critical role in pregnancy and in the newborn. Patients who have had an imminent abortion in their current pregnancy have a significantly higher score of irrationality than those with normal pregnancy, which shows that their emotional state can negatively influence the phenomenon of irrationality. There is a statistically significant association between pregnancy type I (normal pregnancy or imminent pregnancy) and irrationality class (*p* = 0.0001; RR: 2.150, CI (95%): 1.154–4.007). In the case of women with desired pregnancies, the risk of developing irrationality class IV–V is 4.739 times higher, with the association being statistically significant (*p* < 0.0001; RR 4.739; CI (95%): 2.144–10.476). The analysis of the obtained results demonstrates the importance of contributing factors and identifies the possibility of stress disorders, occurring in the last trimester of pregnancy, disorders that can have direct effects on maternal and fetal health. We consider it extremely important to carry out evaluations throughout the pregnancy. At the same time, it is necessary to introduce a screening program to provide psychological counseling in the prenatal care of expectant mothers.

## 1. Introduction

Stress can be described as a process in which “the environment exceeds the ability of an organism to adapt, resulting in psychological and biological changes that can put people at risk of disease”. Stressors can act directly on physiological processes—by releasing stress hormones or by changing immune parameters—without the perception of stress and without arousing negative emotions. The effects of stressors can be mediated by cognitive assessment, followed by suffering if the available coping resources are perceived as inadequate. For women who report negative moods, depressive symptoms, a history of depression, or increased marital conflict, although social support is generally thought to have beneficial effects, some evidence suggests that benefits may not accrue in all circumstances or among all social groups. Social support can cushion the effects of stressors on pregnancy [1].

Psychological stress during pregnancy is defined as “the imbalance that a pregnant woman feels when she cannot cope with the requirements and is expressed both behaviorally and physiologically” [2].

Pregnancy is characterized by changes in neuroendocrine, cardiovascular, and immune function. For this reason, pregnancy itself is perceived as a psychological “stress test”. Research to date has focused on stress exposure. Corroboration of data on stress reactivity and/or severity of stress exposure provided the strongest understanding of the effects of stress on maternal and fetal health [3,4].

According to the literature, the most common stressors that lead to depressive symptoms during pregnancy include: young mother’s age, low socioeconomic status, low educational level (without high school degree), daily stress, and high number of pregnancies [5,6,7].

Data on reactivity to stress during pregnancy are mainly based on studies from the third trimester of pregnancy, and show that both the psychological experience of stress and the impact of psychological activation differ depending on the trimester of pregnancy [8,9,10,11,12].

Prospective studies show that if a pregnant woman is depressed, anxious, or stressed during pregnancy, the baby is more likely to experience a number of adverse neurodevelopmental outcomes, including an increased risk of emotional, behavioral, and cognitive problems [13]. Pregnancy-specific stress has been associated with an increased risk of miscarriage, premature birth, low birth weight, and cesarean delivery [4,14,15,16,17,18].

Pregnant women are more likely to be exposed to physiological stress, such as anxiety about their babies and their completely new lifestyle [19].

The way the pregnancy evolves and all the factors that accompany it, especially the social and cultural dimensions, will rearrange the destiny of the woman in her biological psycho-affective and sociological dimensions [17,20,21].

In 1955, Albert Ellis developed a therapeutic method that he called “rational” because of its emphasis on identifying and modifying the irrational and illogical features of his clients’ thinking. 

According to Ellis’s theory [22], the basis of emotional disorders is the tendency of the individual to make assessments of perceived events, which often take the form of “must-have”, “is mandatory”, and “is absolutely necessary”, and from these, he then derived core irrational beliefs (IB):-Catastrophic beliefs (AWF),-Low tolerance for frustration (LFT),-Global depreciation and valuation (SD/GE).

Considering anxiety as a trigger for stress in pregnant women in the last trimester of pregnancy, we looked at the associated factors that influence the level of stress experienced by pregnant patients.

The aim of the study was to evaluate the influence of associated factors on the level of stress experienced by pregnant patients. We also aimed to identify the population segment with an increased frequency of stress in the last trimester of pregnancy.

## 2. Materials and Methods

### 2.1. Study Design

We conducted a prospective study that included 215 pregnant women in the third trimester of pregnancy, hospitalized in the Obstetrics and Gynecology Clinic II in Târgu-Mureș. The research was conducted between December 2019 and December 2021. All patients included in the study filled in a questionnaire that included 76 questions/items, in which all the data necessary for the study were recorded.

### 2.2. Approval of the Ethics Commisison

The study was conducted according to the guidelines of the Declaration of Helsinki and approved by the Institutional Review Board (or Ethics Committee) of “George Emil Palade” University of Medicine, Pharmacy, Science, and Technology from Târgu-Mureș (No. 199/14/06/2019).

### 2.3. Participants and Procedure

The study included 215 pregnant women in the third trimester of pregnancy, hospitalized in the Obstetrics and Gynecology Clinic II in Târgu-Mureș, with an average age of 25.58 years (SD ± 4.63). 

Using task-specific questionnaires provides a good picture of stress. Psychological stress was assessed using the ABS II questionnaire, validated on the Romanian population [22,23,24]. The questionnaire was distributed by trained staff, with each patient being informed about the importance of the study, its implications, as well as data protection. 

Following the application of inclusion criteria, we took into consideration: age, medium of origin, educational status, marital status, social status, type of pregnancy (normal/imminent abortion), natural delivery, cesarean section, pregnancy age when filling in the questionnaire, if it was a planned pregnancy/desired or an unplanned/unwanted pregnancy, and how many pregnancies (first, second, etc.). As exclusion criteria, we considered the following: pregnancy complications (hypertension, diabetes, bleeding), medical problems during pregnancy (asthma, kidney problems, thyroid problems, intrauterine abnormalities), drug and alcohol use, and severe psychiatric disorders, as well as patients undergoing psychiatric treatment.

Informed consent was obtained from all subjects involved in the study, prior to enrolling in the study.

### 2.4. Measures

Psychological stress can be assessed using validated questionnaires, based on different scales, such as the Attitude and Belief Scale 2 (ABS II) [22,23,24]. The ABS II scale is an important form of evaluation in rational emotional and behavioral therapy [24,25]. According to the theory advanced by Albert Ellis (for details, see [22]), the basis of emotional disorders is the tendency of the individual to make absolutist and rigid assessments of perceived events.

The ABS II scale allows the calculation of scores on different types of irrational beliefs, as well as the estimation of global values of rationality/irrationality. ABS II assesses the irrational and rational beliefs described in Albert Ellis’ theory. The scale was designed by DiGiuseppe, Leaf, Exner, and Robin in 1988 and is a valid measure of the central constructs in rational-emotional and behavioral therapy (REBT) [22,25,26,27]. The efficiency of the instrument is enhanced by the fact that it allows the calculation of separate scores on different types of irrational beliefs, as well as the estimation of global values of rationality/irrationality. Moreover, compared to other clinical trials, it contains a relatively small number of items (72) formulated in accessible language, being easy to administer and quote; when entire administration is not possible, the scale allows the selection of items that assess only a certain type of irrational beliefs, and an individual score can be calculated for them.

Psychometric studies performed on the American population indicate an internal consistency adequate for the use of the instrument; thus, the alpha coefficients for the four processes and three subscales of content vary between 0.92 and 0.86. Most subscales discriminate between clinical and control groups (without psychopathology) [28]. Pilot studies performed on the Romanian population indicate adequate test–retest fidelity and internal consistency: r = 0.7340 (*n* = 80), alpha Cronbach = 0.8654 (*n* = 80) [28,29].

In conclusion, ABS II is one of the most effective tools for assessing irrational/rational beliefs available today.

### 2.5. Statistical Analysis

Data analysis and statistical interpretation included elements of descriptive statistics (frequency, percentage, mean, median, standard deviation) and elements of inferential statistics [30], pp. 155–160. The non-parametric Mann–Whitney test ([30], pp. 76–78) was applied for the comparison of medians, and respectively the Kruskal–Wallis test for 3 or more samples, using the Dunn test for post hoc analysis. For non-Gaussian data (data without normal distribution, data which did not pass the normality test, Shapiro Wilk test), we calculated the Spearman correlation coefficient to measure the correlation between the numerical variables studied. We applied the Chi square test to determine the association between qualitative variables. The significance threshold chosen was 0.05 [30], pp. 174–176. Statistical analysis was performed using GraphPad Prism version 9 utility.

## 3. Results

The study sample included 215 pregnant women in the third trimester of pregnancy, with an average age of 25.58 years (SD ± 4.63), and a mean 36.80-week (SD ± 2.98) pregnancy age at the time of filling in the questionnaire (Table 1).

After analyzing the data on the irrationality scale it was observed that the median scores were higher in patients from rural areas, but no statistically significant difference was observed (*p* = 0.9194) (Table 1).

In terms of marital status, married women obtained an average irrationality score of 138.7 ± 22.02, unmarried women of 133.4 ± 30.09, and divorced women of 135.8 ± 19.52. Applying the Kruskal–Wallis test, we obtained a value of *p* = 0.0124, where most women were married (*n* = 157; 73.02%), compared to those who were divorced (*n* = 32; 14.88%) or unmarried (*n* = 26; 12.09% patients) (Table 1). There was a statistically significant difference between the average irrationality class score for the three types of marital status. 

From the point of view of educational status, the following means of the irrationality score were obtained: high school: 138.8 ± 21.67, middle school: 136.3 ± 23.07, vocational school: 134.6 ± 27.23, post-secondary school: 138.4 ± 22.42, and college: 142.7 ± 10.83 (Table 1). There were no statistically significant differences between types of education.

From the point of view of the type of pregnancy (I/II), the following means of the irrationality score were obtained: type of pregnancy (I): unplanned pregnancy 132.0 ± 28.59 and wanted pregnancy 138.4 ± 21.77, and type of pregnancy (II): normal pregnancy 136.9 ± 23.27 and imminent abortion 146.4 ± 12.30 (Table 1). There were no statistically significant differences between types of pregnancy (I/II).

Depending on the type of birth delivery (I/II), a mean of the irrationality score on delivery was obtained: (I) delivery before term 135.8 ± 26.27 and delivery in term 137.7 ± 22.58, and (II) natural delivery 137.6 ± 22.30 and cesarean section 137.7 ± 23.53. There were no statistically significant differences between types of delivery (I/II).

Regarding the number of pregnancies, the following means of the irrationality score were obtained: one (primiparous) 139.4 ± 19.06, two (multiparous) 136.3 ± 25.58, and more than two (multiparous) 137.6 ± 20.75. There were no statistically significant differences between the three types of pregnancies.

In terms of employment status, the Dunn’s multiple comparison test showed that there was no statistically significant difference between the medians of the irrationality score in subjects in the social status groups (Table 2).

After classifying the irrationality scores, we obtained five classes (Table 3). Most of the subjects included in the study showed high irrationality, and respectively very high irrationality (Table 3). We identified from the study group (215 patients) the frequency and percentage of those with a high score (154; 71.62%) and a very high score (33; 15.35%) of irrationality, which represents an increased risk of developing mental disorders.

We also wanted to identify whether there was a statistically significant association between classes of irrationality and different sociodemographic elements and types of pregnancies (Table 4). Women in urban areas had a 1189 times higher risk of developing class IV–V irrationality than those in rural areas. The association was statistically significant (*p* = 0.0011).

The risk of patients with impending abortion of developing class IV–V irrationality was 2.15 times higher than in patients with normal pregnancy.

There was a statistically significant association between pregnancy type I (normal pregnancy or imminent pregnancy) and irrationality class (*p* = 0.0001; RR: 2.150, CI (95%): 1.154–4.007).

In the case of women with desired pregnancies, the risk of developing irrationality class IV–V was 4.739 times higher, with the association being statistically significant (*p* < 0.0001; RR 4.739; CI (95%): 2.144–10.476).

In terms of delivery, there was no statistically significant association between the natural or cesarean birth and the class of irrationality (Table 4).

## 4. Discussion

In Romania, the hospital provides safety and indirectly a reduction of stress.

The aim of this study was to analyze the factors that determine the appearance of stress in the last (third) trimester of pregnancy. The third trimester of pregnancy is a critical time, as many physical and emotional changes occur before birth. Studies have been performed that evaluated the occurrence of stress in all trimesters of pregnancy, demonstrating variations in stress. Weng’s study in China showed that stress symptoms were more prevalent in the last trimester of pregnancy [31].

Traumatic events can trigger a variety of psychological responses that can have an impact on well-being over time. High levels of psychological distress are common and may signal a level of psychological arousal that helps to explain the link between trauma and long-term health problems [32].

The results obtained from the study showed that pregnant women in urban areas are more vulnerable than those in rural areas. After analyzing the data on the irrationality scale it was observed that the median scores were higher in patients from rural areas, but no statistically significant difference was observed (*p* = 0.9194). From a sociological point of view, there are differences between the two rural/urban environments, considering the village/city opposition—differences between two opposing typological worlds. The village as a community can provide peace of mind, while the city can generate stress. 

Marital status is also an important sociodemographic factor in mental suffering, especially in the case of single people or divorced people. Existing vulnerabilities that precede pregnancy can interact with marital status, increasing stress levels, producing effects on the maternal–fetal system. Family can have major implications for pregnancy through counseling, support, prenatal care, and emotional security stability [33]. Our results indicated a statistically significant difference between the average irrationality class score for the three types of marital status, and divorced patients had a statistically significantly higher average irrationality score, and thus divorce can be a predictive factor for mental disorders. Specialist studies suggest a possible relationship between marital satisfaction and lifestyle. Pregnant women who have increased marital satisfaction have a much lower level of stress. A poor marital relationship is the most stable predictor of anxiety and other health problems during pregnancy [34]. The result of the non-parametric Mann–Whitney test indicates a statistically significant difference between the average irrationality class score for the three types of marital status, with divorced patients having statistically significantly increased irrationality scores.

Education or the educational level of the mother is another factor that influences mental suffering. The workplace is a strong predictor of increased mental suffering and can lead to a number of related exposures, including a range of potential stress variables. A low level of education is directly related to socioeconomic disadvantages, because patients fear that they will not be able to meet the needs of the child [35,36]. In our study, the irrationality scores were not statistically significantly correlated with the level of education. 

In our study group, patients with a college level of education (subject with bachelor degree) were few, followed by graduates of post-secondary school, secondary education, vocational school, and high school. Stress was associated with a lower level of education, and this was consistent with the results of specialized studies [37,38]. 

Low socioeconomic status and insecurity are accompanied by increased stress, as evidenced by studies showing that housewives, either employed in the private sector or for an indefinite period of time, had a higher level of mental distress than women who have not experienced this stressor [39]. The results obtained in our study confirmed that social status can have an impact on the evolution of stress in pregnant women in the last trimester of pregnancy, but this impact was not statistically significant. Thus, housewives and those employed in the private sector/for indefinite periods had a high level of stress, due to living conditions, busy schedule, and work responsibilities. 

According to literature, women who come from unstable social backgrounds, and without the support of their partner, are not financially prepared to cope with the demands of life [40,41], and carry a heavy emotional burden due to unwanted pregnancies [21,36]. 

In our study, pregnant women who did not plan their pregnancy, compared to those who had desired pregnancies, had scores of irrationality classes in grades I–III, indicating a *p* < 0.0001, considered statistically significant, which means that patients with a planned pregnancy had a higher score of irrationality, theoretically being exposed to developing mental disorders.

In terms of pregnancy type I, we found a statistically significant association between pregnancy type I (normal pregnancy or imminent pregnancy) and irrationality class (*p* = 0.0001; RR: 2.150, CI (95%): 1.154–4.007).

Lobel et al. showed that the frequency of premature births is based on chronic stress, which can increase the risk of having a low birth weight [10]. 

Our findings show that patients who gave birth at the expected time (at 9 months of pregnancy) had a lower class of irrationality compared to those who gave birth prematurely, but without statistical significance (RR: 0.9712; 95% (CI): 0.7656–1.232; *p* = 0.6796).

Additionally, from the point of view of pregnancy type (cesarean section and natural delivery), patients who gave birth naturally had a higher risk of irrationality score (RR: 0.9712, 95% (CI): 0.7656–1.232, *p* = 0.6796), but without statistical significance.

Pregnant women who wanted to give birth by cesarean section chosed this method of delivery due to fear of pain and desire for comfort, reasons that can cause a high degree of stress, while patients who knew the benefits of a natural birth wanted a normal birth. Prenatal stress causes an increased risk of conduct disorders and cognitive problems, and more research is needed to detect and treat emotional problems and mental disorders that occur in the last trimester of pregnancy [42].

Although some psychological factors have been associated with healthy behaviors, few studies have explored the relationship of several psychosocial factors with the lifestyle of pregnant women [34]. 

The stressors that commonly affect women during pregnancy around the world are low material resources, unfavorable living conditions, family responsibilities, and pregnancy complications. Social support is a vital resource for pregnant women, which positively influences pregnancy and protects from possible mental illnesses.

Our study has some limitations that deserve attention in the future, such as the relatively small number of patients included in the study. It would be useful for future studies to be conducted on a larger sample of subjects, beginning with the first trimester of pregnancy. The second limitation is represented by the validity of the instrument, which is an ongoing process, and the clinical expertise of the investigator, who administred the test to the subjects, has definite importance.

## 5. Conclusions

The analysis of the results obtained based on sociodemographic indicators demonstrated the importance of contributing factors and identified the possibility of stress disorders, occurring in the last trimester of pregnancy, disorders that can have direct effects on maternal and fetal health. Family support, marital harmony, and childbirth education for pregnant women are also factors that need to be improved to help women cope with the changes and challenges of pregnancy and childbirth, all of which contribute to increasing the quality of life.

High and very high scores of irrationality were associated with an increased risk of developing mental disorders.

We consider it extremely important to carry out evaluations throughout the pregnancy. 

At the same time, it is necessary to introduce a screening program to provide psychological counseling in the prenatal care of expectant mothers, especially those with a predisposition to stress and anxiety, mothers from disadvantaged social backgrounds, and those with low social status. 

Pregnant women with high and very high scores of irrationality should be included in a program for the prevention of mental disorders.

## Figures and Tables

**Table 1 ijerph-19-08315-t001:** Mean ± SD, median and *p*-values of irrationality score *.

	The Study Sample (215)	Mean ± Standard Deviation	Median	*p*-Value
Age	Age (years)	25.5 ± 4.63	25.0	
	Gestational age when filling in the questionnaire (weeks)	36.8 ± 2.98	38.0	
Place of origin	Rural area	136.3 ± 25.64	141.5	* 0.9194
	Urban area	138.7 ± 19.96	143.0	
Civil status	Married	138.7 ± 22.02	143.0	* 0.0124
	Divorced	135.8 ± 19.52	138.0	
	Not married	133.4 ± 30.09	140.5	
Education	Middle school	136.3 ± 23.07	141.0	* 0.7922
	Highschool	138.8 ± 21.67	142.5	
	Vocational school	134.6 ± 27.23	140.0	
	Post-secondary school	138.4 ± 22.42	143.5	
	College	142.7 ± 10.83	144.0	
Type of pregnancy (I)	Unplanned pregnancy	132.0 ± 28.59	139.0	* 0.0212
	Wanted pregnancy	138.4 ± 21.77	143.0	
Type of pregnancy (II)	Normal pregnancy	136.9 ± 23.27	142.0	* 0.0442
	Imminent abortion	146.4 ± 12.30	151.0	
Delivery type (I)	Delivery before term	135.8 ± 26.27	142.0	* 0.8884
	Delivery in term	137.7 ± 22.58	142.0	
Delivery type (II)	Natural delivery	137.6 ± 22.30	142.0	* 0.9715
	Cesarean section	137.7 ± 23.53	142.5	
Number of pregnancies	One (primiparous)	139.4 ± 19.06	142.0	* 0.9999
	Two (multiparous)	136.3 ± 25.58	142.5	
	More than two (multiparous)	137.6 ± 20.75	143.0	

* The non-parametric Mann–Whitney test was applied to compare the means.

**Table 2 ijerph-19-08315-t002:** *p*-values for irrationality scale according to social status *.

Dunn’s Multiple Comparison Test	*p*-Value
Housewife vs. State employee	0.3825
Housewife vs. Private employee	0.7507
Housewife vs. Determined period employee	0.2450
Housewife vs. Employed for indefinite period	0.9772
State employee vs. Private employee	0.5365
State employee vs. Determined period employee	0.4151
State employee vs. Employed for indefinite period	0.4754
Private employee vs. Determined period employee	0.2842
Private employee vs. Employed for indefinite period	0.7383
Determined period employee vs. Employed for indefinite period	0.4012

* To compare medians, the Kruskal–Walls test was applied to 3 or more samples, using the Dunn test for post hoc analysis.

**Table 3 ijerph-19-08315-t003:** Frequency and percentage of subjects included in the study according to irrationality classes.

Study Sample (215)	Frequency	Percentage
Very low irrationality (I)	4	1.86%
Low irrationality (II)	9	4.19%
Medium level irrationality (III)	15	6.98%
High irrationality (IV)	154	71.62%
Very high irrationality (V)	33	15.35%

**Table 4 ijerph-19-08315-t004:** Irrationality class IV–V vs. irrationality class I–III *.

Study Sample (215)		Irrationality Class IV–V	Irrationality Class I–III	*p-*Value
Place of origin	Rural area	85	22	RR: 0.1189 CI (95%): 1.069–1.323 *p* = 0.0011
	Urban area	102	6	
Pregnancy type I	Normal pregnancy	172	28	RR: 2.150 CI (95%): 1.154–4.007 *p* = 0.0001
	Imminent abortion	6	9	
Pregnancy type II	Unplanned pregnancy	5	22	RR: 4.739 CI (95%): 2.144–10.476 *p* < 0.0001
	Wanted pregnancy	165	23	
Pregnancy type I	Delivery before term	11	2	RR: 0.9712 CI (95%): 0.7656–1.232 *p* = 0.6796
	In term delivery	176	26	
Pregnancy type II	Cesarean section	80	8	RR: 1.079 CI (95%): 0.9762–1.193 *p* = 0.2159
	Natural delivery	107	20	

* We applied the Chi square test to determine the association between qualitative variables (binary).

## Data Availability

The data presented in this study are available on request from the corresponding author.

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
