# Peer review of "Psychological Stress Perceived by Pregnant Women in the Last Trimester of Pregnancy"

_ijerph, 2022, doi:10.3390/ijerph19148315_

Round 1
Reviewer 1 Report
-The authors should indicate the authors of the scales and whether the scales are validated in Romania.
-The psychometric properties of the scales used are not indicated.
-The timing of sample collection includes biases such as covid-19 pandemic, collection, before and during the pandemic
-It is not necessary to explain how a quotation of the scale is inverted, it is indicated in the psychometric properties.
-The statistical assumptions achieved are not clear-Table 1 indicates the p-value but not the type of test used. In addition, the effect size and confidence intervals are not indicated. Similar for Table 2 as a complementary analysis.
-Table 3 seems to indicate the analysis of the anchor points of the questionnaire. Normally this type of analysis is performed according to factor and not according to item.
-Table 4 presents p values and it is difficult to understand the test under study, moreover, it does not speak of mean differences, it speaks of relationships.
The methodology and results are not very comprehensible and transparent, so the authors should reformulate them almost completely.
Author Response
Dear Reviewer,
Thank you very much for reading my article! I answered to all your observations as follws:
The authors should indicate the authors of the scales and whether the scales are validated in Romania.
We mention that in Romania, the hospital provides safety and indirectly a reduction of stress.
The psychometric properties of the scales used are not indicated.
Using task-specific questionnaires provides a good picture of stress. Psychological stress can be assessed using validated questionnaires, based on different scales, such as Attitude and Belief Scale 2 - ABS II [22–24]. The ABS II scale is an important form of evaluation in rational emotional and behavioral therapy [24,25]. According to the theory advanced by Albert Ellis (for details see Ellis & Dryden, 1997), the basis of emotional disorders is the tendency of the individual to make absolutist and rigid assessments of perceived events.
The ABS II scale allows the calculation of scores on different types of irrational beliefs, as well as the estimation of global values of rationality / irrationality. ABS II assesses the irrational and rational beliefs described in Albert Ellis' theory. The scale was designed by DiGiuseppe, Leaf, Exner and Robin in 1988 and is a valid measure of the central constructs in rational-emotional and behavioral therapy (REBT) [22, 25-27]. The efficiency of the instrument is enhanced by the fact that it allows the calculation of separate scores on different types of irrational beliefs, as well as the estimation of global values of rationality / irrationality. Moreover, compared to other clinical trials, it contains a relatively small number of items (72) formulated in accessible language, being easy to administer and quote; when entire administration is not possible, the scale allows the selection of items that assess only a certain type of irrational beliefs, and an individual score can be calculated for them.
Psychometric studies performed on the American population indicate an internal consistency adequate to the use of the instrument; thus the alpha coefficients for the four processes and three subscales of content vary between .92 and .86. Most subscales discriminate between clinical and control groups (without psychopathology) [28] Pilot studies performed on the Romanian population indicate adequate test-retest fidelity and internal consistency: r = .7340 (N = 80); alpha Cronbach = .8654 (N = 80) [28, 29]
In conclusion, ABS II is one of the most effective tools for assessing irrational / rational beliefs available today.
The timing of sample collection includes biases such as covid-19 pandemic, collection, before and during the pandemic
The project was approved before the Covid 19 pandemic, and the questionnaire and project was approved by the Ethics Committee in June 2019.
It is not necessary to explain how a quotation of the scale is inverted, it is indicated in the psychometric properties.
We deleted that part from the article, as it was not necesarry.
The statistical assumptions achieved are not clear-Table 1 indicates the p-value but not the type of test used. In addition, the effect size and confidence intervals are not indicated. Similar for Table 2 as a complementary analysis.
Table 1. Mean+SD, median and p-values of irrationality score.*
*The non-parametric Mann Whitney test was applied to compare the means
Table 2. p values for irrationality scale according to social status.*
*To compare medians, the Kruskal-Walls test was applied to 3 or more samples, using the Dunn test for post-hoc analysis.
-Table 3 seems to indicate the analysis of the anchor points of the questionnaire. Normally this type of analysis is performed according to factor and not according to item.
After classifying the irrationality scores, we obtained 5 classes. (Table 3). Most of the subjects included in the study showed high irrationality, respectively very high irrationality (Table 3). We identified from the study group (215 patients) the frequency and percentage of the segment with high score (154; 71.62%) and very high score (33; 15.35%) of irrationality, which represents an increased risk of developing mental disorders.
-Table 4 presents p values and it is difficult to understand the test under study, moreover, it does not speak of mean differences, it speaks of relationships.
We deleted the table, after re-reading the article.
The methodology and results are not very comprehensible and transparent, so the authors should reformulate them almost completely.
We remade and reformulated the part including methodology, discussions and results.
We conducted a prospective study that included 215 pregnant women in the third trimester of pregnancy, hospitalized in the Obstetrics and Gynecology Clinic II in Târgu-MureÈ™. The research was conducted between December 2019 and December 2021.All patients included in the study filled in a questionnaire that included a number of 76 questions / items, in which all the data necessary for the study were recorded.
The study included 215 pregnant women in the third trimester of pregnancy, hospitalized in the Obstetrics and Gynecology Clinic II in Târgu-MureÈ™, with an average age 25.58 years (SD ± 4.63).
Using task-specific questionnaires provides a good picture of stress. Psychological stress was assessed using the ABS II questionnaire, validated on the Romanian population[22–24]. The questionnaire was distributed by the trained staff, each patient being informed about the importance of the study, its implications, as well as data protection.
Following the application of inclusion criteria, we took into consideration: age, medium of origin, educational status, marital status, social status, type of pregnancy (normal / imminent abortion), natural delivery, cesarean section, pregnancy age when filling in the questionnaire, if it was a planned pregnancy / desired or an unplanned / unwanted pregnancy, how many pregnancies there are (first, second, etc.), and as exclusion criteria, we considered the following: pregnancy complications (hypertension, diabetes, bleeding), medical problems during pregnancy (asthma, kidney problems, thyroid problems, intrauterine abnormalities), drug and alcohol use, and severe psychiatric disorders, as well as patients undergoing psychiatric treatment
Informed consent was obtained from all subjects involved in the study, prior to enrolling in the study.
Using task-specific questionnaires provides a good picture of stress. Psychological stress can be assessed using validated questionnaires, based on different scales, such as Attitude and Belief Scale 2 - ABS II [22–24]. The ABS II scale is an important form of evaluation in rational emotional and behavioral therapy [24,25]. According to the theory advanced by Albert Ellis (for details see Ellis & Dryden, 1997), the basis of emotional disorders is the tendency of the individual to make absolutist and rigid assessments of perceived events.
The ABS II scale allows the calculation of scores on different types of irrational beliefs, as well as the estimation of global values of rationality / irrationality. ABS II assesses the irrational and rational beliefs described in Albert Ellis' theory. The scale was designed by DiGiuseppe, Leaf, Exner and Robin in 1988 and is a valid measure of the central constructs in rational-emotional and behavioral therapy (REBT) [22, 25-27]. The efficiency of the instrument is enhanced by the fact that it allows the calculation of separate scores on different types of irrational beliefs, as well as the estimation of global values of rationality / irrationality. Moreover, compared to other clinical trials, it contains a relatively small number of items (72) formulated in accessible language, being easy to administer and quote; when entire administration is not possible, the scale allows the selection of items that assess only a certain type of irrational beliefs, and an individual score can be calculated for them.
Psychometric studies performed on the American population indicate an internal consistency adequate to the use of the instrument; thus the alpha coefficients for the four processes and three subscales of content vary between .92 and .86. Most subscales discriminate between clinical and control groups (without psychopathology) [28] Pilot studies performed on the Romanian population indicate adequate test-retest fidelity and internal consistency: r = .7340 (N = 80); alpha Cronbach = .8654 (N = 80) [28, 29]
In conclusion, ABS II is one of the most effective tools for assessing irrational / rational beliefs available today.

Reviewer 2 Report
I have read this paper with great interest, value the data, but there is still quite some work to make the data and interpretation clearer to the readership.
Title
Why not add ‘reported’ or ‘perceived’ psychological stress in the title ?
Abstract
If possible (word count) the magnitude of differences, and its direction (rural versus city, ‘different’) should be provided in the abstract section of the paper.
In the abstract, you mention ‘hospitalized’, does this truly means admitted in the hospital, as (the need for) admission itself will likely affect the perceived/reported stress ? If it are truly hospitalisations, and assuming that the hospital is in city, distance form home, or clinical indication might be the determinant ?
Questionnaire is mentioned, but nothing on its validity ?
Was parity relevant ?
How do you explain the link with stress and type of parity ? where these ‘planned’ type of partus already known to the mother, or was this a post stress assessment observation ?
Methods
The total number of questionnaires is a result, and we also need some idea on the number of cases considered/included, and preferably, also an idea on similarities between women who provided consent, and those who preferred not to be involved;
The scale rather assesses (ir)rationality, but I could not retrieve the stress assessment ? this should be better described.
Result
How should I understand immanent abortion, as the mean gestational age is about 36 weeks ?
Again, it is not clear to me how ‘stress’ has been assessed ?
Author Response
Dear Reviewer,
Thank you very much for reading my article! I answered to all your observations as follws:
I have read this paper with great interest, value the data, but there is still quite some work to make the data and interpretation clearer to the readership.
Title
Why not add ‘reported’ or ‘perceived’ psychological stress in the title ?
Psychological stress perceived by pregnant women in the last trimester of pregnancy
Abstract
If possible (word count) the magnitude of differences, and its direction (rural versus city, ‘different’) should be provided in the abstract section of the paper.
In the abstract, you mention ‘hospitalized’, does this truly means admitted in the hospital, as (the need for) admission itself will likely affect the perceived/reported stress ? If it are truly hospitalisations, and assuming that the hospital is in city, distance form home, or clinical indication might be the determinant ?
Abstract: Pregnancy is characterized by changes in neuroendocrine, cardiovascular and immune function. For this reason, pregnancy itself is perceived as a psychological "stress test". Research to date has focused on stress exposure. The aim of the study was to evaluate the influence of associated factors on the level of stress experienced by pregnant patients. We conducted a prospective study that included 215 pregnant women in the third trimester of pregnancy, hospitalized in the Obstetrics and Gynecology Clinic II in Târgu-MureÈ™, between December 2019 - December 2021, who were evaluated by ABS II scale. All patients included in the study, filled in a questionnaire that included a number of 76 questions / items, in which all the data necessary for the study were recorded. The results obtained from the study showed that pregnant women in urban areas (53.49%) are more vulnerable than those in rural areas (46.51%), being influenced by social and professional stressors, social determinants playing a critical role in pregnancy and in the newborn. Patients who have had an imminent abortion in their current pregnancy have a significantly higher score of irrationality than those with normal pregnancy, which shows that their emotional state can negatively influence the phenomenon of irrationality There is a statistically significant association between pregnancy type I (normal pregnancy or imminent pregnancy) and irrationality class (p=0.0001; RR: 2,150, CI (95%): 1,154-4.007). In the case of women with desired pregnancies, the risk of developing irrationality class IV-V is 4,739 times higher, the association being statistically significant (p <0.0001; RR 4,739; CI (95%): 2,144-10,476). The analysis of the results obtained, demonstrates the importance of contributing factors and identifies the possibility of stress disorders, occurring during pregnancy in the last trimester of pregnancy, disorders that can have direct effects on maternal and fetal health. We consider it extremely important to carry out evaluations throughout the pregnancy. At the same time, it is necessary to introduce a screening program to provide psychological counseling in the prenatal care of expectant mothers.
Questionnaire is mentioned, but nothing on its validity ?
2.4.Measures
Using task-specific questionnaires provides a good picture of stress. Psychological stress can be assessed using validated questionnaires, based on different scales, such as Attitude and Belief Scale 2 - ABS II [22–24]. The ABS II scale is an important form of evaluation in rational emotional and behavioral therapy [24,25]. According to the theory advanced by Albert Ellis (for details see Ellis & Dryden, 1997), the basis of emotional disorders is the tendency of the individual to make absolutist and rigid assessments of perceived events.
The ABS II scale allows the calculation of scores on different types of irrational beliefs, as well as the estimation of global values of rationality / irrationality. ABS II assesses the irrational and rational beliefs described in Albert Ellis' theory. The scale was designed by DiGiuseppe, Leaf, Exner and Robin in 1988 and is a valid measure of the central constructs in rational-emotional and behavioral therapy (REBT) [22, 25-27]. The efficiency of the instrument is enhanced by the fact that it allows the calculation of separate scores on different types of irrational beliefs, as well as the estimation of global values of rationality / irrationality. Moreover, compared to other clinical trials, it contains a relatively small number of items (72) formulated in accessible language, being easy to administer and quote; when entire administration is not possible, the scale allows the selection of items that assess only a certain type of irrational beliefs, and an individual score can be calculated for them.
Psychometric studies performed on the American population indicate an internal consistency adequate to the use of the instrument; thus the alpha coefficients for the four processes and three subscales of content vary between .92 and .86. Most subscales discriminate between clinical and control groups (without psychopathology) [28] Pilot studies performed on the Romanian population indicate adequate test-retest fidelity and internal consistency: r = .7340 (N = 80); alpha Cronbach = .8654 (N = 80) [28, 29]
In conclusion, ABS II is one of the most effective tools for assessing irrational / rational beliefs available today.
Was parity relevant ?
How do you explain the link with stress and type of parity ? where these ‘planned’ type of partus already known to the mother, or was this a post stress assessment observation ?
In our study, pregnant women did not plan their pregnancy, compared to those who were desired pregnancies, have scores of irrationality classes in grades I-III, indicating a p <0.0001, considered statistically significant, which means that patients with a planned pregnancy have a higher score of irrationality, theoretically being exposed to developing mental disorders.
In terms of pregnancy type I, we found a statistically significant association between pregnancy type I (normal pregnancy or imminent pregnancy) and irrationality class (p=0.0001; RR: 2,150, CI (95%): 1,154-4.007).
Methods
The total number of questionnaires is a result, and we also need some idea on the number of cases considered/included, and preferably, also an idea on similarities between women who provided consent, and those who preferred not to be involved;
215 patientes were include in the study, as they gave their consent for being included in the study and fulfilled the inclusion criteria.
The scale rather assesses (ir) rationality, but I could not retrieve the stress assessment ? this should be better described.
The results obtained from the study showed that pregnant women in urban areas are more vulnerable than those in rural areas. After analyzing the data on the irrationality scale, it was observed that the median scores were higher in patients from rural, but no statistically significant difference was observed (p = 0.9194). From a sociological point of view, there are differences between the 2 rural / urban environments, even talking about the village / city opposition, differences as between two opposing typological worlds. The village as a community can provide peace of mind, while the city can generate stress.
Marital status is also an important sociodemographic factor in the mental suffering, especially in the case of single people or divorced people. Existing vulnerabilities that precede pregnancy can interact with marital status, increasing stress levels, producing effects on the maternal-fetal system. Family can have major implications for pregnancy through counseling, support, prenatal care, emotional security stability [35]. Our results indicated statistically significant difference between the average irrationality class score for the three types of marital status, divorced patients with a statistically significantly higher average irrationality score, so divorce can be a predictive factor for mental disorders. Specialist studies suggest a possible relationship between marital satisfaction and lifestyle. Pregnant women who have increased marital satisfaction have a much lower level of stress. A poor marital relationship is the most stable predictor of anxiety and other health problems during pregnancy [36]. The result of the Mann Whitney nonparametric test indicates a statistically significant difference between the average irrationality class score for the three types of marital status, with divorced patients having statistically significantly increased irrationality scores.
Education or the educational level of the mother is another factor that influences the mental suffering. Workplace is a strong predictor of increased mental suffering and can lead to a number of related exposures, including a range of stress potential variables. Low level of education that is directly related to socio-economic disadvantages, because patients fear that they will not be able to meet the needs of the child [39,40]. In our study, the irrationality scores were not statistically significantly correlated with the level of education.
In our study group, patients with college studies was small; followed by graduates of post-secondary school, secondary education, vocational school and high school. Stress was associated with a lower level of education, and was consistent with the results of specialized studies. [37,38].
Low socioeconomic status and insecurity are accompanied by increased stress, as evidenced by studies showing that housewives, either employed in the private sector or for an indefinite period of time, had a higher level of mental distress than women who have not experienced this stressor [41]. The results obtained in our study confirmed that social status can have an impact on the evolution of stress in pregnant women in the last trimester of pregnancy, but not statistically significant. Thus, housewives and employed in private / indefinite periods, had a high level of stress, due to living conditions, busy schedule and work responsibilities.
In according to Bouchard G, Garfin D.R and Thompson O and co., women who come from unstable social backgrounds, without the support of their partner, are not financially prepared to cope with the demands of life [42,43], carry a heavy emotional burden due to unwanted pregnancies [21,40].
In our study, pregnant women did not plan their pregnancy, compared to those who were desired pregnancies, have scores of irrationality classes in grades I-III, indicating a p <0.0001, considered statistically significant, which means that patients with a planned pregnancy have a higher score of irrationality, theoretically being exposed to developing mental disorders.
In terms of pregnancy type I, we found a statistically significant association between pregnancy type I (normal pregnancy or imminent pregnancy) and irrationality class (p=0.0001; RR: 2,150, CI (95%): 1,154-4.007).
Lobel M et al, showed that the frequency of premature births is based on chronic stress, which can increase the risk of having a low birth weight. [10]
Our findings show that patients who gave birth prematurely have a lower class of irrationality compared to those who gave birth prematurely but without statistical significance (RR: 0.9712; 95% CI): 0.7656-1.232; p = 0.6796)
Also, from the point of view of pregnancy type (cesarean section and natural delivery), patients who gave birth naturally had a higher risk of irrationality score (RR: 0.9712, 95% CI): 0.7656-1.232, p = 0.6796) but without statistical significance.
Pregnant women who wanted to give birth by cesarean section, invoked the fear of pain, comfort, reasons that can cause a high degree of stress, and patients who knew the benefits of natural birth and wanted a normal birth. Prenatal stress causes an increased risk of conduct disorders and cognitive problems, so more research is needed to detect and treat emotional problems and mental disorders that occur in the last trimester of pregnancy [44].
Although some psychological factors have been associated with healthy behaviors, few studies have explored the relationship of several psychosocial factors with the lifestyle of pregnant women [36].
The stressors that commonly affect women during pregnancy around the world are low material resources, unfavorable living conditions, family responsibilities and pregnancy complications. Social support is a vital resource for pregnant women, which positively influences pregnancy and protects them from possible mental illness [45].
Our study has some limitations that deserve attention in the future. The relatively small number of patients included in the study.Future studies would be useful to be conducted on a larger sample of subjects, beginning with the first trimester of pregnancy. The second limitation is represented by the validity of an instrument, which is an ongoing process, the clinical expertise of the test user has its definite importance.
Result
How should I understand immanent abortion, as the mean gestational age is about 36 weeks ?
In our study, pregnant women did not plan their pregnancy, compared to those who were desired pregnancies, have scores of irrationality classes in grades I-III, indicating a p <0.0001, considered statistically significant, which means that patients with a planned pregnancy have a higher score of irrationality, theoretically being exposed to developing mental disorders.
In terms of pregnancy type I, we found a statistically significant association between pregnancy type I (normal pregnancy or imminent pregnancy) and irrationality class (p=0.0001; RR: 2,150, CI (95%): 1,154-4.007).
Lobel M et al, showed that the frequency of premature births is based on chronic stress, which can increase the risk of having a low birth weight. [10]
Our findings show that patients who gave birth prematurely have a lower class of irrationality compared to those who gave birth prematurely but without statistical significance (RR: 0.9712; 95% CI): 0.7656-1.232; p = 0.6796)
Again, it is not clear to me how ‘stress’ has been assessed ?
High and very high scores of irrationality are associated with an increased risk of developing mental disorders.
We consider it extremely important to carry out evaluations throughout the pregnancy.
At the same time, it is necessary to introduce a screening program to provide psychological counseling in the prenatal care of expectant mothers, especially those with a predisposition to stress and anxiety, mothers from disadvantaged social backgrounds, and with low social status.
Pregnant women with high and very high scores of irrationality should be included in a program for the prevention of mental disorders.

Round 2
Reviewer 2 Report
suggest to accept